# Cellular and Molecular Genetic Mechanisms of Lung Fibrosis Development and the Role of Vitamin D: A Review

**DOI:** 10.3390/ijms25168946

**Published:** 2024-08-16

**Authors:** Darya Enzel, Maxim Kriventsov, Tatiana Sataieva, Veronika Malygina

**Affiliations:** Medical Institute Named after S.I. Georgievsky, V.I. Vernadsky Crimean Federal University, Lenina Boulevard 5/7, 295051 Simferopol, Russia; darya.enzel@yandex.ru (D.E.); maksimkgmu@mail.ru (M.K.); vera.maligina@mail.ru (V.M.)

**Keywords:** lungs, fibrosis, inflammation, vitamin D, vitamin D receptors

## Abstract

Idiopathic pulmonary fibrosis remains a relevant problem of the healthcare system with an unfavorable prognosis for patients due to progressive fibrous remodeling of the pulmonary parenchyma. Starting with the damage of the epithelial lining of alveoli, pulmonary fibrosis is implemented through a cascade of complex mechanisms, the crucial of which is the TGF-β/SMAD-mediated pathway, involving various cell populations. Considering that a number of the available drugs (pirfenidone and nintedanib) have only limited effectiveness in slowing the progression of fibrosis, the search and justification of new approaches aimed at regulating the immune response, cellular aging processes, programmed cell death, and transdifferentiation of cell populations remains relevant. This literature review presents the key modern concepts concerning molecular genetics and cellular mechanisms of lung fibrosis development, based mainly on in vitro and in vivo studies in experimental models of bleomycin-induced pulmonary fibrosis, as well as the latest data on metabolic features, potential targets, and effects of vitamin D and its metabolites.

## 1. Introduction

Pulmonary fibrosis (idiopathic pulmonary fibrosis) is a stage-by-stage, progressive, and irreversible lesion of the lung tissue with increasing respiratory failure and an unfavorable prognosis [1]. At the same time, the etiopathogenesis of this disease remains insufficiently studied. Consequently, the available therapeutic approaches are also significantly limited, with the presence of only one approach that has proven effective in relation to patient survival—lung transplantation [2]. The median life expectancy of patients with idiopathic pulmonary fibrosis is only 2–3 years [3], which indicates an aggressive and extremely progressive lesion of the respiratory parts of the lungs, with a constant and increasing burden on the public healthcare system from year to year. In particular, based on the US Medicare Database, Raghu et al. reported an increased cumulative prevalence of idiopathic pulmonary fibrosis (IPF) from 202.2 cases per 100,000 persons in 2001 to 494.5 cases per 100,000 persons in 2011 [4]. In a special high-risk population (among a national cohort of US veterans), the prevalence of IPF increased from 276 cases per 100,000 in 2010 to 725 cases per 100,000 in 2019 [5]. Generally, pulmonary fibrosis is considered a rare disease, with adjusted prevalence estimates (per 10,000 of the population) for each country ranging from 0.57 to 4.51 in Asia–Pacific countries, 0.33 to 2.51 in Europe, and 2.40 to 2.98 in North America [6]. However, epidemiologic data related to other countries, including Africa, South America, South Asia, and the Middle East regions, are limited. Based on one of the largest multinational observational longitudinal registry of IPF patients (European MultiPartner IPF Registry) (*n* = 1620), idiopathic pulmonary fibrosis can be considered predominantly a disease of older people (mean age at diagnosis 67.6 years), with more males (71%) and smokers (63% with smoking history) affected [7].

Taking into account the above-mentioned epidemiological data and rising burden, the search for new highly effective and safe approaches to pharmacotherapy for this pathological condition, designed to improve the quality of life of patients with the restoration of the structure of lung tissue, or at least, slowing down the rate of progression of fibrotic changes, is still very important. Some of these approaches to therapy were the use of immunosuppressants (glucocorticoid therapy, azathioprine, cyclophosphamide, etc.) [8], anticoagulants [9], endothelin receptor antagonists (bosentan, ambrisentan, and macitentan) [10], and various antifibrotic and immunomodulatory drugs (interferon gamma, etanercept, imatinib, etc.) [11,12], which, in general, have proven ineffective in randomized phase II or III clinical trials. At the same time, currently, inhibition of transforming growth factor beta (TGF-β) is considered one of the promising approaches, whose crucial role in the development of pulmonary fibrosis has been repeatedly confirmed, both in experimental and clinical studies [13], as well as the suppression of some other growth factors, including vascular endothelial growth factors (VEGFR-1, VEGFR-2, and VEGFR-3), fibroblast growth factors (FGFR-1, FGFR-2, and FGFR-3), and platelet growth factors (PDGFR-α and PDGFR-β) (Table 1) [14]. In particular, pirfenidone (an inhibitor TGF-β) [15] and nintedanib (an inhibitor of vascular endothelial growth factor receptors, platelet growth factor, and fibroblast growth factor) [16] are the only medicines that are officially approved for treatment of idiopathic pulmonary fibrosis. At the same time, despite a slight slowdown in fibrotic changes in the lungs and an improvement in the quality of life of patients, these drugs are still unable to change the overall progression of the disease and high mortality rates during the first 3–5 years from the moment of diagnosis [17]. In this regard, a detailed study of the pathogenesis of the disease and the search for possible treatment methods with pathogenetically justified targets continues, with the continued interest of researchers in nature-based therapeutic approaches, among which one of the key places is occupied by vitamin D and its analogs with effects realized through ligand-associated activation of vitamin D receptors.

This review attempts to systematize the data available in the literature on key factors and features of the pathogenesis of pulmonary fibrosis in terms of potential targets of effects of ligand-associated activation of vitamin D receptors, as well as to present existing experimental and clinical data on the effects of vitamin D and its analogs in conditions of progressive pulmonary fibrosis.

## 2. Key Pathogenetic Mechanisms and Cell Populations in the Development of Pulmonary Fibrosis

Understanding the key pathogenetic mechanisms, including the various cell subpopulations and signaling pathways involved in the progression of fibrotic transformation of the pulmonary parenchyma, seems to be one of the key tasks that researchers face in order to identify informative prognostic markers and develop new modern approaches to the treatment of patients with pulmonary fibrosis.

Pulmonary fibrosis is a complex and multi-component pathological process of unexplained etiology, probably associated with the chronic exposure of damaging factors of various natures and origins to the respiratory tract of the lungs [25]. Starting with the damage to the alveolar epithelium, followed by a cascade of coagulation reactions, inflammatory response, and regeneration with remodeling of lung tissue, pulmonary fibrosis appears to be a pathological process with dysregulation of reparative regeneration and excessive progressive growth of fibrous connective tissue. As with regard to etiology, although assumptions are made about the role of key risk factors such as smoking, exposure to ionizing radiation, etc. [26,27], potential molecular mechanisms that underlie in the development of these disorders remain not fully clarified. As an illustration of our lack of understanding of the role of certain factors in the development of pulmonary fibrosis, population-based studies on tobacco smoking can be taken as an example, the results of which are often multidirectional [28,29]. At the same time, the role of inflammation and its proliferative phase with a shift in the system of intercellular regulation towards proliferation, activation, and differentiation of collagen-synthesizing fibroblastic cells is obvious, including due to the mechanism of the epithelial–mesenchymal transition with excessive deposition of extracellular matrix components [30].

According to classical concepts, pulmonary fibrosis is a stage-by-stage process with a sequential transition from the phase of damage to the phase of inflammatory response, followed by reparative regeneration and remodeling of lung tissue. Genetic factors, including various single-nucleotide polymorphisms of genes (in particular, in the gene of a protein interacting with Toll-like receptors, which is a suppressor of the TGF-β signaling pathway), are currently considered as initial predisposing factors [31,32], the effect of environmental factors (such as smoking and various bacterial, viral, or fungal infections) [28,29,33], age-related changes with increased secretion of proinflammatory cytokines, and antiapoptotic effects on fibroblastic cells [34]. In this regard, both the effect of external damaging factors and age-related changes are accompanied by a number of epigenetic modifications, including DNA methylation and various histone modifications, as well as changes in the expression of various non-coding RNAs (microRNAs) [35]. In particular, an increase in expression of miRNA-21 and miRNA-199 with a decrease in the expression level of miRNA-31 and miRNA-200 was demonstrated against the background of idiopathic pulmonary fibrosis [36,37]. These epigenetic factors may play a decisive role in starting a cascade of pathological reactions. In particular, miRNA-21 has been shown to be able to induce epithelial–mesenchymal transition and promote the development of TGF-β-mediated fibrosis [38], whereas miRNA-200, on the contrary, stimulates the process of transdifferentiation into type I alveolar epithelium [39]. Among other epigenetic factors that could potentially be involved in the regulation of lung tissue remodeling, other small non-coding RNAs have also been identified, including miRNA-145, miRNA-424, miRNA-301a, miRNA-29, and others [40]. The data obtained in the framework of these studies, among other things, open up new prospects for therapeutic approaches based on blocking specific miRNAs [41].

Starting with damage to the alveolar epithelium and vascular endothelium of the microcirculatory bed with a proinflammatory response and activation of the coagulation cascade, the pathological process enters a phase of progression with predominantly TGF-β-mediated cell activation of fibroblastic series and formation of typical foci of fibrosis. At the same time, all major cell populations are involved in the formation of pulmonary fibrosis, including alveolocytes of type I and II, cells of the stromal microenvironment (fibroblasts, myofibroblasts, and vascular endothelium), and inflammatory cells (including neutrophils, monocytes/macrophages, and lymphoid cells).

### 2.1. Alveolar Epithelium

According to most researchers, damage to the alveolar epithelium is an invariable trigger point in the development of idiopathic pulmonary fibrosis. Type I alveolar epithelium lines more than 90% of the alveolar surface, and its damage triggers the activation program of type II alveolocytes, which are surfactant-producing cells, as well as progenitor cells under conditions of reparative regeneration [42]. Under normal conditions, damage to type I alveolocytes followed by type II alveolocyte hyperplasia is completed by the restoration of the epithelial lining of the alveoli due to transdifferentiation of type II alveolocytes into type I alveolar epithelium [43]. At the same time, this transdifferentiation from type II alveolocytes to type I alveolocytes is carried out through an intermediate cellular form—a population of so-called basaloid cells. In vivo and in vitro studies have demonstrated that effects on this intermediate cell population can both stimulate and inhibit their further differentiation into type I alveolocytes. It is also shown that in the foci of fibrosis in patients with idiopathic pulmonary fibrosis, the number of these cells is significantly increased [44], and that this cell population is able to transdifferentiate into keratin-5-positive basal cells (CK5+) with subsequent differentiation into bronchial epithelium, which may explain the phenomenon of bronchialization of alveoli in idiopathic pulmonary fibrosis [45]. In addition, another type of transdifferentiation of type II alveolocytes is likely due to activation of epithelial–mesenchymal transition mechanisms [46]. In general, in the case of damage to the basement membrane, as well as against the background of a pronounced increase in vascular permeability due to damage to the endothelium, reepithelization does not occur, and type II alveolocytes continue to secrete numerous growth factors and proinflammatory and profibrotic cytokines. At the same time, the alveolar epithelium is the main cell population responsible for the production of platelet growth factor (PDGF), transforming growth factor beta (TGF-β), and tumor necrosis factor alpha (TNF-α), which are key orchestrators in the system of intercellular interaction and the subsequent development of pulmonary fibrosis [42]. A combination of external damaging factors and internal factors, such as cell aging with telomere shortening, mitochondrial dysfunction, endoplasmic reticulum stress, disorders of apoptosis, and autophagy processes, lead to ineffective restoration of damaged epithelium due to type II alveolocytes. Ultimately, this leads to the activation and proliferation of fibroblasts and myofibroblasts and excessive deposition of extracellular matrix components.

### 2.2. Cells of the Stromal Microenvironment—Fibroblasts, Myofibroblasts, and Vascular Endothelium

After damage to the alveolar epithelium with a violation of reparative regeneration processes and the formation of proinflammatory and profibrotic cytokine background, fibroblasts are activated and their transdifferentiation into myofibroblasts that actively secrete collagen [47]. Under normal conditions, a resolution phase occurs in which myofibroblasts are deactivated, fibroblast recruitment is suppressed, and cellular detritus along with excess amounts of extracellular matrix are resorbed by macrophages. In the case of idiopathic pulmonary fibrosis, this process is disrupted by a central mechanism of TGF-β-mediated signaling pathway. Meanwhile, the expression of TGF-β is observed in almost all cell populations, including alveolar epithelium (including in aberrant basaloid cells), alveolar macrophages, fibroblastic cells, and cells of the immune system [48]. Under these conditions, the transdifferentiation of fibroblasts into myofibroblasts with increased expression of alpha-smooth muscle actin (α-SMA) creates conditions for active collagen synthesis and excessive accumulation of extracellular matrix [49]. Differentiation of myofibroblasts and inhibition of apoptosis in this cell population is supported by many factors, including TGF-β, integrin aVβ6, PDGF, VEGF, various coagulation factors, Wnt-mediated signaling pathways, etc. [50,51]. Similar factors lead to an increase in the expression of mesenchymal and corresponding suppression of the expression of epithelial markers, which act as fundamentals of the implementation of the epithelial–mesenchymal transition. Usually, the epithelial–mesenchymal transition is characterized by the loss of epithelial markers (for example, E-cadherin) and the appearance of expression of mesenchymal markers (N-cadherin, vimentin, or α-SMA). In conditions of idiopathic pulmonary fibrosis, the population of fibroblastic cells is not the result of epithelial transdifferentiation. In this case, the epithelial–mesenchymal transition leads to the expression of mesenchymal markers in the alveolar epithelium, which contributes to the progression of fibrosis [52].

Endothelial cells are also involved in the pathogenesis of idiopathic pulmonary fibrosis, being an integral part of stromal cell microenvironments. Firstly, damage to the endothelium in the respiratory parts of the lungs leads to the activation of the coagulation cascade that, along with other cytokines released by the alveolar epithelium, contributes to the development of an inflammatory response [53,54]. In the areas of connective tissue proliferation in pulmonary fibrosis, there is an increase in the expression of VEGF and IL-8, both in endothelial cells and in type II alveolocytes [55]. Meanwhile, VEGF is one of the key mediators of angiogenesis, suppressing endothelial apoptosis and stimulating its proliferation and differentiation. Also, similar to the epithelial–mesenchymal transition, there is experimental data indicating the ability of endothelial cells to transdifferentiate into fibroblasts and myofibroblasts, with activation of this mechanism due to TGF-β and Ras/MAPK signaling pathways [56].

### 2.3. Immunocompetent Cells

Neutrophils, being a cell population that is one of the first to respond to damage to the alveolar epithelium, according to most researchers, are not key effectors of a dysregulatory inflammatory response and progressive fibrosis. However, neutrophils are able to secrete a number of profibrotic factors (in particular, IL-8/CXCL8), metalloproteinases, and take an active part in the remodeling of the extracellular matrix [57,58]. In addition, neutrophils are capable of forming so-called extracellular neutrophil traps, the appearance of which is associated with the stimulation of fibroblasts and the induction of fibrosis [59].

Unlike neutrophils, macrophages represent one of the key cell populations involved in the implementation of the resolving phase of inflammation and subsequent reparative regeneration. Various subpopulations of macrophages in the foci of damage are able to secrete a number of cytokines (including IL-1, IL-6, TNF-α, TGF-β, various metalloproteinases, and growth factors), thereby regulating activation cells of the stromal microenvironment, the degree of deposition of extracellular matrix, and angiogenesis processes [60]. Meanwhile, the emphasis in studying the population of monocyte–macrophage cells, whether alveolar or interstitial macrophages, is to identify two key subpopulations of cells: proinflammatory macrophages type 1 (M1) with the classical activation pathway and anti-inflammatory macrophages type 2 (M2) with an alternative activation pathway [61], although these two subpopulations themselves are not similar [62]. The induction of M1 macrophages is associated with the action of bacterial lipopolysaccharides and TNF-α, whereas the M2 subpopulation responds to IL-4 and IL-10 stimuli and is actively involved in the resolving phase of the inflammatory process and the occurrence of fibrosis [63]. A number of studies have shown that in conditions of pulmonary fibrosis, M2 macrophages predominate, actively secreting such profibrotic growth factors as TGF-β, fibroblast growth factor, PDGFa, IGF-1, and VEGF, stimulating excessive deposition of extracellular matrix [64,65]. This is also confirmed in experimental studies with bleomycin-induced pulmonary fibrosis, in which an increase in the number of subpopulations of macrophages M2 was observed on the background of increased expression of IL-1β with subsequent transdifferentiation of type II alveolocytes into type I alveolocytes [66]. Thus, M2 macrophages are the key orchestrators of the dysregulated regeneration of the alveolar epithelium and the launch of a cascade of profibrotic tissue remodeling in conditions of progressive pulmonary fibrosis.

Like macrophages, T-cells also appear to be one of the key cellular subpopulations involved in the implementation of the inflammatory response and subsequent fibrosis, primarily due to modulation of the immune response. The role of various subpopulations of T-helpers has been studied quite well, in particular, their distribution into type 1 T-helpers (Th1) and type 2 T-helpers (Th2). Th1 cells are inducers of a proinflammatory response, secreting a number of cytokines (for example, IFN-γ and IL-12) and contributing to the realization of the exudative phase of inflammation [67], whereas Th2 cells contribute to the development of proliferative reactions in the foci of inflammation by secreting IL-4, IL-5, IL-9, IL-13, and a number of other cytokines [68]. It has been established that such profibrotic cytokines as IL-13 and IL-4 contribute to the differentiation and activation of myofibroblasts [69]. Thus, it can be considered that the shift towards Th2 cells is one of the key pathogenetic links in progressive pulmonary fibrosis. The start of this cellular mechanism of adaptive immune response is apparently triggered by damaged epithelial cells and macrophages by active secretion of TGF-β, IL-1β, and other cytokines, followed by recruitment of T-cells. At the same time, the role of other T-cell subpopulations is also being studied in the development of pulmonary fibrosis, in particular T-regulatory cells, Th17, Th9, and γδT cells, for which multidirectional modulating effects have been demonstrated [70,71].

In particular, the cascade of IL-17-mediated immune reactions is currently considered as one of the central mechanisms in the pathogenesis of pulmonary fibrosis from the point of view of intercellular interactions [72,73]. Although the IL-17 family (IL-17A–IL-17F) plays an important role in the formation of an anti-infectious immune response, dysregulation of the expression of individual representatives of this family of cytokines (mainly IL-17A) is one of the main factors, triggering profibrotic changes and contributing to their progression [74]. Although it is currently established that the secretion cytokines of the IL-17 family can be carried out by epithelial cells, dendritic cells, macrophages, and various subpopulations of lymphocytes (including γδT-cells, cytotoxic CD8+ T-cells, etc.), the key cellular subpopulation remains specialized T-helpers of the 17th type (Th17), differentiation which is carried out from naive CD4+ cells in conditions of cytokine co-stimulation due to IL-1β, TGF-β, IL-6, and IL-23 [75]. In experimental studies on a model of bleomycin-induced pulmonary fibrosis, it was shown that IL-17A is secreted by Th17 and acts as one of the inducers and factors of progression of fibrous remodeling of the pulmonary parenchyma [76]. Similar data on the role of the Th17-dependent immune response were obtained for IL-1β-induced pulmonary fibrosis, as well as pulmonary fibrosis induced by the graft-versus-host reaction [77,78].

Some significant interactions between vitamin D/VDR and different subpopulations of T-cells are shown in Figure 1.

Likewise, the role of the B-cell link in implementation of the profibrotic tissue response is actively discussed. In particular, under conditions of pulmonary fibrosis development, activation of B-cells with an increase in the release of a number of cytokines and metalloproteinases has been demonstrated, which contributes to dysregulation of the resolving phase of inflammation and excessive deposition of extracellular matrix [79,80,81].

### 2.4. Cell Aging and Apoptosis

Despite advances in understanding the key pathogenetic mechanisms that lead to the development of a dysregulated immune response in lung parenchyma, followed by progressive fibrosis, the trigger mechanisms are still poorly understood. Currently, as such, a combination of factors includes cell aging (primarily, the epithelial lining of the alveoli), as well as a violation of the regulation of the processes of programmed cell death (apoptosis).

In this aspect, cellular aging is a complex regulated process, including the so-called replicative aging and stress-induced premature aging of cells, caused in two ways by both genetic and epigenetic factors and damaging environmental factors [82], accompanied by typical changes in the form of telomere shortening, DNA damage and epigenetic modifications, mitochondrial dysfunction, and oxidative stress [83,84]. Both the age-related physiological and pathologically conditioned process of cellular aging are accompanied by the formation of the so-called senescence-associated secretory phenotype (SASP), characterized, among other things, by an increase in the secretion of cytokines such as IL-1, IL-6, TGF-β, TNF-α, MMP-2, and MMP-9 [85]. In recent studies, IL-11, which is part of the IL-6 family and is actively secreted by fibroblasts in response to profibrotic stimulation, including TGF-β, IL-13, and fibroblast growth factors, has also been assigned a key role [86,87].

At the same time, the role of cellular aging as a trigger mechanism extends to various cell populations, including alveolar epithelial cells, fibroblasts, and endothelium, although the key importance, as before, is assigned to the aging of alveolocytes type II and basaloid cells [88] with corresponding changes in their genetic apparatus (including, among other things, an increase in the expression of plasminogen activator inhibitor 1 (PAI-1) and the p53 signaling pathway) [89]. The role of cellular aging in microenvironment cells with increased expression of alpha-smooth muscle actin (α-SMA) and increased secretion of extracellular matrix in fibroblasts has also been established [90]. According to Yanai et al., fibroblasts in conditions of pulmonary fibrosis show pronounced signs of replicative cellular aging, but unlike epithelial cells, they are resistant to oxidative stress and apoptosis [91]. Clinical observations confirm the role of cellular aging. In particular, it has been shown that age is one of the most significant risk factors, with a twofold increase in the frequency of development of pulmonary fibrosis for every decade over the age of 50 [92], consistent with the theory of cellular aging. Moreover, both in experimental models [93,94] and in pilot clinical studies [95], it has been demonstrated that the elimination of cellular subpopulations with signs of aging or therapy aimed at leveling the effects of cellular aging (in particular, with usage of a combination of dasatinib and quartzetine) [96] leads to a significant slowdown in the progression of pulmonary fibrosis and restoration of lost function.

Invariably related to the issue of cellular aging and renewal of cellular subpopulation, the role of programmed cell death (apoptosis) in the conditions of development of pulmonary fibrosis is also important. At the same time, according to in vitro and in vivo studies, changes in the regulation of apoptosis in various cell populations are often multidirectional [97,98]. In particular, alveolocytes in conditions of pulmonary fibrosis development are characterized by a significantly higher level of readiness for apoptotic death, manifested, among other things, by a significant increase in the expression of the proapoptotic protein p53 [99]. Hypoxia-induced endoplasmic stress, TGF-β/p38 MAPK-mediated signaling pathway, angiotensin receptor signaling pathway, and classical Fas-mediated mechanism are considered as possible mechanisms and signaling pathways of apoptosis activation in this cell population [100]. Unlike type II alveolocytes, fibroblasts and myofibroblasts in conditions of pulmonary fibrosis development are less sensitive to oxidative stress and, on the contrary, exhibit increased resistance to apoptosis [101,102]. Meanwhile, these multidirectional changes in alveolocytes and cells of mesenchymal origin in conditions of pulmonary fibrosis in both cases are associated with cellular aging and changes in gene expression, including PAI-1 and p53 [103], although detailed mechanisms of acquisition by various cellular subpopulations of different sensitivity to the action of proapoptotic signals still remain unexplored. Among others, there is evidence indicating the role of various epigenetic mechanisms, in particular, miRNA-34a [104,105].

## 3. Effects of Vitamin D and Its Analogues Implemented through Ligand-Associated Activation of Vitamin D Receptors

Although there are currently a number of treatment approaches for the patients with pulmonary fibrosis that have proven, albeit limited, effectiveness, there is still an unmet need for the development of additional measures, including drugs with immunomodulatory effects. From this point of view, vitamin D, its endogenous derivatives (including CYP11A1-mediated derivatives), as well as a number of artificially synthesized selective VDR agonists, showed promising results in various preclinical studies and can act as promising correction agents aimed at key pathogenetic mechanisms of fibrotic transformation.

Vitamin D is currently considered a biologically active substance, which has a wide range of effects that are not limited to the regulation of calcium and phosphorus homeostasis [106]. The so-called “non-classical” effects of vitamin D, which include regulation of cell proliferation and differentiation, apoptosis, intercellular adhesion, oxidative stress, and inflammatory responses, are of particular interest to researchers [107]. Entering the body from the outside with food or synthesized endogenously in the skin under the action of UV irradiation, vitamin D is metabolized into its active form (1,25-dihydroxy vitamin D) due to specific enzymes-hydroxylases (25-hydroxylase and 1α-hydroxylase), which exhibit the full range of its biological effects through interaction with the vitamin D receptor (VDR), which is a ligand-dependent transcription factor [108]. At the same time, the possibility of metabolic formation of an active form of vitamin D is not limited to the liver and kidneys, and currently both specific hydroxylases and VDR expression have been detected in almost all tissues and cell populations, including epithelial alveolar cells, alveolar macrophages [109,110], and immune cells [111], which emphasizes the important physiological role of vitamin D in helping to ensure tissue homeostasis and the functioning of the immune system due to autocrine and paracrine effects. Relatively recently, an alternative pathway of vitamin D metabolism has also been identified with the participation of an enzyme of the cytochrome P450—CYP11A1 family with the formation of numerous sets of hydroxyl derivatives, among which 20(OH)D3, 20,23(OH)2D3, 1,20(OH)2D3, 1,20,23(OH)3D3, and 17,20,23(OH)3D3 are of higher importance, all of which are not inferior but often surpass the classical active form of vitamin D, 1,25(OH)2D3, in terms of physiological activity while having unexpressed or no calcemic effects at all [112,113]. As well as the enzymes of the classical metabolic pathway, the expression of CYP11A1 was detected in many cells, including immune cells [114]. Similar to 1,25(OH)2D3, CYP11A1-mediated derivatives are able to bind to a specific VDR [115], as well as, equally important, to other nuclear receptors, including retinoic acid receptors (RXR), liver X-receptors (LXR), and other receptors representing transcription factors, which expands the potential range of effects of these ligands [116].

Currently, available data from in vitro and in vivo studies on various cell populations, including immunocompetent cells, allow us to consider vitamin D, its endogenous metabolites (including CYP11A1-mediated ones), and a number of its synthetic analogs, that are the agonists of vitamin D receptors (paricalcitol), as promising agents for the prevention and treatment of pulmonary fibrosis.

The potential antifibrotic mechanisms of ligand-associated activation of vitamin D receptors are based on various effects on key signaling pathways, including SMAD [117], p38 MAPK [118], NF-kB [119], JAK/STAT [120], PPAR-α/γ [121,122], and calcineurin/NFAT [123]. In addition to this, ligand-dependent transcriptional effects of VDR on components of other signaling pathways, including Nrf2 [124], are explained. As one of the mechanisms mediated by some of the above signaling pathways and implemented under conditions of fibrous remodeling of the pulmonary parenchyma, the mechanism of VDR-mediated inhibition of the renin–angiotensin system (RAS) by suppressing renin expression, on the one hand, and inhibition of TGF-β/SMAD and p38 MAPK signaling pathways, on the other hand, is described [125]. It is shown that the RAS components (mainly angiotensin II) stimulate the formation of extracellular matrix and increase the expression of TGF-β, contributing to the development of pulmonary fibrosis [126]. In general, the vitamin D/VDR complex, acting as a transcription factor through various application points, controls the processes of cellular differentiation and proliferation and is also involved in the regulation of the immune response (Table 2).

In terms of interaction with the above-mentioned key signaling pathways, in relation to the pathogenetic links in the development of pulmonary fibrosis, a number of immunotropic effects of ligand-associated activation of VDR have been demonstrated in studies. In general, vitamin D (ligand-associated activation of VDR) is assigned a suppressor function with the redistribution of T helper subpopulations from Th1 cells towards Th2 and Treg. It is assumed that vitamin D can play a dual role: along with the late effects of stimulating the resolving phase of inflammation, the effects of the initial activation of the inflammatory response are also manifested [127,128]. Meanwhile, the value of vitamin D as an antifibrotic agent in conditions of pulmonary fibrosis from this point of view may be ambiguous, given one of the key roles of Th2 cells in the implementation of the profibrotic response [129]. On the other hand, a study by Joshi S et al. demonstrated a transcription-mediated effect of 1,25(OH)2D3 with a decrease in IL-17A expression by blocking (NFAT), suppression of histone deacetylase, and transcription factor Runx1 with an increase in Foxp3 expression and a shift in the Th17/Treg ratio towards immunosuppressive T-regulatory cells [130].

Given the key role of the alveolar epithelium in triggering the cascade of pulmonary parenchyma remodeling, the data obtained using alveolocyte cell cultures are of particular interest. In a study by Heejae Han et al., obesity in laboratory animals was associated with the development of vitamin D deficiency, accompanied by increased expression of TGF-β, components of the renin–angiotensin system, and proinflammatory cytokines (IL-1β, IL-6, and TNF-a). At the same time, the addition of vitamin D to the culture of human bronchial epithelial cells and alveolocytes of laboratory mice significantly reduced the expression of TGF-β1 [131], which can be regarded as one of the key effects from a pathogenetic point of view in terms of preventing progressive fibrosis. In an in vitro study by Allan M. Ramirez et al., a dose-dependent suppression of the key profibrotic cytokine expression (TGF-β1) has also been demonstrated in the culture as fibroblasts and epithelial cells of human lung tissue under conditions of application of 1,25(OH)2D3 [132]. In vitro experiments also found that vitamin D significantly suppressed expression of the PSAT1 gene and activation of the MAPK-mediated signaling pathway in human pulmonary fibroblast culture [118]. In addition, it was found that an increase in the expression of vitamin D receptors in fibroblasts in conditions of pulmonary fibrosis is, apparently, one of the protective and adaptive mechanisms that limit the proliferation and activation of fibroblastic cells by suppressing the JAK1/STAT3 signaling pathway [133].

However, the data available in the literature regarding the effects of vitamin D are contradictory. In particular, in a study by Trinidad Guijarro et al. (2018) using representative cell models of type II alveolocytes and myofibroblasts in vitro, as well as a model of bleomycin-induced pulmonary fibrosis in vivo, it was shown that the use of 1,25(OH)2D3 induced cellular aging and aggravated histopathological changes in the lungs [134]. The authors associated the obtained results with a potential violation of DNA repair in the presence of vitamin D in conditions of bleomycin-induced damage. At the same time, a later study by the same research group confirmed the absence of similar negative effects in in vitro and in vivo models for hypocalcemic vitamin D analogs (paricalcitol and calcipotriol), making those preferred candidates for leveling bleomycin-induced damage [135]. At the same time, it is worth noting that there is practically no data on CYP11A1-mediated vitamin D derivatives in relation to pulmonary fibrosis.

With reference to the epithelial–mesenchymal transition in conditions of pulmonary fibrosis, in the study by Fei Jiang et al., it was shown that incubation of human alveolar epithelial cells (A549) with TGF-β (5 ng/mL) led to an increase in the expression of mesenchymal markers (N-cadherin and vimentin) in them with a corresponding suppression of the expression of epithelial markers (E-cadherin). Meanwhile, the addition of 1α.25-dihydroxy vitamin D3 (50 nmol) to the cell culture prevented these changes with suppression of the expression of transcription factors associated with epithelial–mesenchymal transition (Snail and β-catenin) and normalization of mRNA levels of fibronectin and type I collagen genes [136]. Similar effects of vitamin D related to suppression of epithelial–mesenchymal transition have also been demonstrated in another study of alveolar cell culture [137].

Experimental models of pulmonary fibrosis in rodents, mainly using intratracheal instillation [138,139] or aerosol inhalation of bleomycin [140,141], have introduced significant understanding of the cellular and molecular genetic mechanisms of pulmonary fibrosis and potential application points of vitamin D and its analogs. In particular, in an experimental study on rodents, it was found that the use of vitamin D contributed to a significant decrease in the manifestations of bleomycin-induced fibrosis with a decrease in hydroxyproline levels, expression of smooth muscle actin and TGF-β, and the severity of histopathological and ultrastructural signs of lung tissue damage [142]. Application of vitamin D in conditions of bleomycin-induced pulmonary fibrosis contributed to a decrease in the expression levels of mRNA collagen type I, type III, and smooth muscle actin with the restoration of the bleomycin-induced decrease in VDR mRNA expression. Suppression of phosphorylation of TGF-β1-induced SMAD was one of the mechanisms of antifibrotic effects of vitamin D [143].

In another study using an experimental model of bleomycin-induced pulmonary fibrosis in C57BL/6 mice, it was shown that the use of a vitamin D receptor agonist (paricalcitol) significantly prevented body weight loss and contributed to less pronounced fibrotic changes in lung tissue, whereas vitamin D deficiency, on the contrary, was accompanied by significantly more pronounced damage to the lung parenchyma. The use of paricalcitol led to suppression of the expression of TGF-β, α-SMA, type I collagen, and fibronectin, as well as components of the renin–angiotensin system (angiotensinogen, angiotensin II, and type I angiotensin receptors) [125].

Key mechanisms of pulmonary fibrosis development and potential application points of vitamin D and its metabolites, which implement their effects through VDR-dependent modulation of signaling pathways, are schematically presented in Figure 2 based on in vitro and in vivo research data.

In contrast to experimental studies in vitro and in vivo, data from clinical trials, including multicenter randomized trials, are extremely heterogeneous and, for the most part, indicate a lack of evidence regarding the role of vitamin D and its potential effects on the risk of developing and progression of pulmonary fibrosis.

In particular, a recent large-scale study using Mendelian randomization revealed the absence of any significant links between genetically determined levels of circulating vitamin D (25(OH)D) and the risk of developing idiopathic pulmonary fibrosis [144]. Although this study showed no significant association between vitamin D levels and the development of pulmonary fibrosis, it is not possible to completely disregard the possibility of diet–gene or gene–environment interactions influencing such findings. Given the extensive metabolic pathway of vitamin D, it seems that the functionality of other organs (like the liver and kidneys) may play a significant role in the status of vitamin D and its metabolites, affecting the evaluation of causal associations. Moreover, there is evidence that different individuals may display a different molecular and biochemical response to the same dose of either long-term or single-bolus vitamin D3 supplementation. As a part of the proposed hypothesis of the acquired vitamin D resistance, it was shown that approximately one-quarter of individuals in the general population are low responders and are not able to exert the expected vitamin D-regulatory effects [145,146]. These factors may act as significant limitations to the reliable assessment of the results of population studies, and further both experimental and large-scale clinical studies with proper stratification are needed to explore this issue. Meanwhile, a number of other studies have established a relationship between vitamin D deficiency and the development of a number of pulmonary pathologies, including respiratory infectious diseases and chronic obstructive pulmonary disease [147,148]. In another clinical study involving a few subjects, it was demonstrated that the use of a combination of vitamins D, C, and E in patients with idiopathic pulmonary fibrosis leads to an improvement in respiratory function and a decrease in inflammatory response and oxidative stress [149].

## 4. Conclusions

The data obtained so far regarding the nature and key mechanisms of the development of pulmonary fibrosis suggest significant progress in understanding the pathogenesis of fibrous remodeling of the pulmonary parenchyma. Starting with damage reactions, the pathological process is realized through a cascade of complex molecular genetic mechanisms, the key to which is TGF-β/SMAD, an indirect pathway involving various cell populations, including alveolocytes of type I and II, cells of the mesenchymal microenvironment such as fibroblasts and myofibroblasts, antigen-presenting cells, and a wide range of T-cell subpopulations. In this regard, it is the dysregulatory immune response with a profibrotic inflammatory response vector that is the key target of both existing and potential therapeutic approaches aimed at leveling the fibrous transformation of the pulmonary parenchyma, restoration of respiratory function, and improvement of survival rates and quality of life in patients with pulmonary fibrosis.

From the point of view of regulating the immune response, based on the data of in vitro and in vivo studies, the use of vitamin D and its metabolites seems promising (including CYP11A1-mediated derivatives), which realize their effects through VDR-dependent modulation of the signaling pathways TGF-β/SMAD, p38 MAPK, NF-kB, JAK/STAT, PPAR-α/γ, NFAT, Nrf2, and others, manifested, among other things, in a change in the distribution key cellular subpopulations, cytokine profile, the processes of cellular aging, and programmed cell death. Meanwhile, the absence of significant evidence to date of the effectiveness of vitamin D and its derivatives in idiopathic pulmonary fibrosis in clinical studies may be associated with both genetic polymorphism, epigenetic modifications, dietary and lifestyle features on the one hand, and possible imperfection of existing generally accepted models of bleomycin-induced pulmonary fibrosis on the other hand. However, accumulated preclinical data and the understanding of key application points of VDR-mediated effects in remodeling conditions of pulmonary parenchyma emphasize the need for further study of this issue and resolution of the problem of inconsistency between preclinical and clinical data.

## Figures and Tables

**Figure 1 ijms-25-08946-f001:**
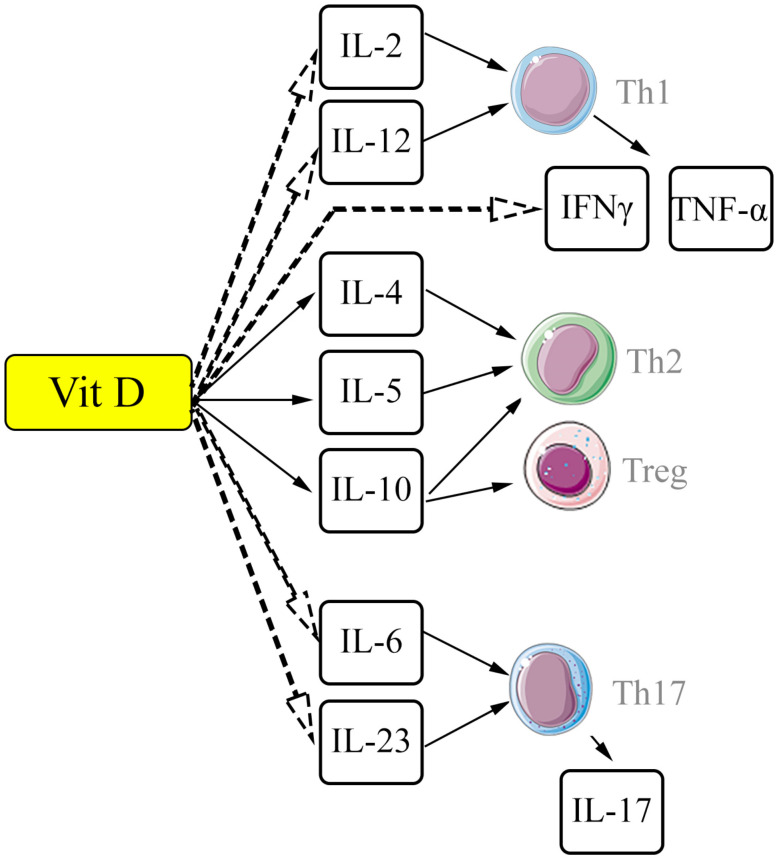
Diagram of some interactions between vitamin D and T cells. Vitamin D and its metabolites suppress cytokines related to proinflammatory Th1 and Th17 immune responses but stimulate secretion of cytokines associated with Th2 and Treg. The scheme was generated using images from Servier Medical Art. Servier Medical Art by Servier is licensed under a Creative Commons Attribution 3.0 Unported License (https://creativecommons.org/licenses/by/3.0/), accessed on 10 July 2024 Abbreviations: Vit D—vitamin D and its metabolites; IL—interleukins; Th1—T-helper type 1; Th2—T-helper type 2; Th17—T-helper type 17; Treg—regulatory T-cell; IFNγ—interferon gamma; TNF-α—tumor necrosis factor alpha. Solid arrows—stimulation/enhancing effects; dashed arrows—inhibitory effects.

**Figure 2 ijms-25-08946-f002:**
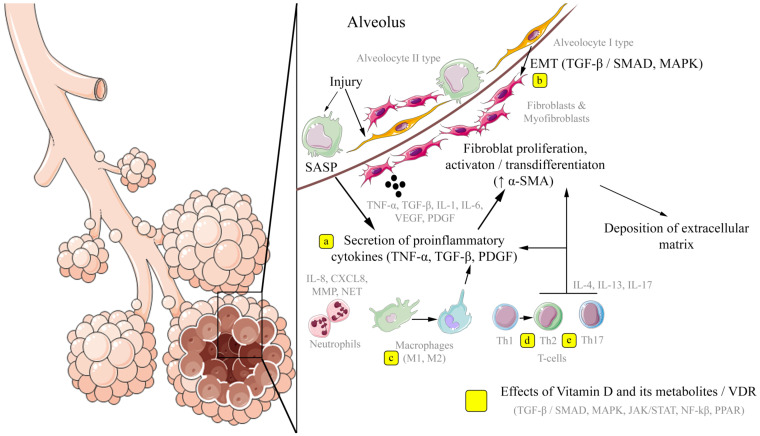
Diagram of the key mechanisms of pulmonary fibrosis development and potential application points for the effects of vitamin D and its metabolites. Initiated by damage to alveolocytes, pulmonary fibrosis is mediated by a proinflammatory response with dysregulatory excess extracellular matrix deposition. Possible effects of vitamin D and its metabolites/VDR may include inhibition of the secretion of proinflammatory cytokines (IL-1β and TGF-β), including by suppressing the renin–angiotensin system (**a**); suppression of the epithelial–mesenchymal transition (**b**); modulation of the polarization of macrophages on M1 and M2 subpopulations (**c**); an increase in the subpopulation of Th2 cells and a shift from Th17 towards T-regulatory cells (**d**,**e**). The scheme was generated using images from Servier Medical Art. Servier Medical Art by Servier is licensed under a Creative Commons Attribution 3.0 Unported License (https://creativecommons.org/licenses/by/3.0/), accessed on 10 July 2024. Abbreviations: SASP (senescence-associated secretory phenotype)—cellular phenotype associated with aging; EMT—epithelial–mesenchymal transition; TNF-a—tumor necrosis factor alpha; TGF-β—transforming growth factor beta; PDGF—platelet growth factor; a-SMA—smooth muscle actin alpha; IL—interleukins; MMP—metalloproteinases; NET—neutrophil extracellular trap; Th1—T-helper type 1; Th2—T-helpers type 2; Th17—T-helpers type 17; VDR—vitamin D receptor.

**Table 1 ijms-25-08946-t001:** List of some clinical trials evaluating inhibitors of TGF-β and other growth factors in patients with idiopathic pulmonary fibrosis.

Drug	Target	Study Design, Phase, and Number of Participants	Summary of Main Findings	Name and Status of Clinical Trial (NCT)
Pirfenidone	TGF-β	Randomized, double-blind, placebo-controlled, parallel-group study; Phase 3; in total, 779 patients with IPF	Positive dose-dependent effect with reduced decline in FVC vs. placebo and favorable benefit risk profile [18]	CAPACITY, completed (NCT00287729, NCT00287716)
		Randomized, double-blind, placebo-controlled, parallel-group study; Phase 3; 555 patients with IPF	Reduced disease progression, as reflected by lung function, exercise tolerance, and progression-free survival [19]	ASCEND, completed (NCT01366209)
		Open-label extension study; Phase 3; 1058 patients with IPF	Mean change in percent predicted FVC from baseline at 180 weeks was −9.6%; median on-treatment survival from the first pirfenidone dose was 77.2 months [20]	RECAP, completed (NCT00662038)
BG00011 (anti-αvβ6 monoclonal antibody)	TGF-β/TGF-β receptors—integrins αV/β6	Randomized, double-blind, placebo-controlled, parallel-group study; Phase 2; 109 patients with IPF	Early trial termination due to imbalance in adverse events and lack of clinical benefit [21]	SPIRIT, terminated (NCT03573505)
PLN-74809 (Bexotegrast)	TGF-β/TGF-β receptors—integrins αV/β1, αV/β6	Randomized, double-blind, dose-ranging, placebo-controlled, parallel-group study; Phase 2a; 120 patients with IPF	Dose-dependent antifibrotic effect (reduction in FVC decline over 12 weeks vs. placebo) [22]	INTEGRIS-IPF, completed (NCT04396756)
Nintedanib	Tyrosine kinases (VEGFR 1–3, FGFR 1–3, PDGFR α, β)	Randomized, double-blind, dose-ranging, placebo-controlled, parallel-group study; Phase 3; 663 patients with progressive fibrosing interstitial lung disease	Slower rate of progression of the interstitial lung disease (reduction in FVC decline) [23]	INBUILD, completed (NCT02999178)
Pamrevlumab	CTGF	Randomized, double-blind, placebo-controlled, parallel-group study; Phase 3; in total, 728 patients with IPF	Termination of a planned open-label extension study of pamrevlumab as well as the ongoing ZEPHYRUS-2 trial due to a lack of effectiveness [24]	ZEPHYRUS-1,2, terminated (NCT03955146, NCT04419558)

**Table 2 ijms-25-08946-t002:** Some key signaling pathways and effects of ligand-associated activation of vitamin D receptors.

Signaling Pathway	Signaling Pathway Inductor	The Implemented Cascade of Reactions	Effect of Vit D Complex/VDR
SMAD	TGF-β/receptors TGF-β; angiotensin II	Regulation of cellular cycle, differentiation (in particular of myofibroblasts), immune reactions	Suppression by reducing TGF-β expression and nuclear translocation of SMAD components
MARK	TGF-β, growth factors, lipopolysaccharides, etc.	Regulation of cellular proliferation and differentiation, inflammatory response, and elimination of cells by apoptosis	Suppression by enhancing the expression of proteinase (MAPK phosphatase-1) followed by inhibition of p38 MAPK
NF-κβ	Proinflammatory cytokines (IL-1 and TNF-α), lipopolysaccharides, and growth factors	Regulation of non-specific and adaptive immunity and inflammatory response	Suppression by inhibiting Ikkß kinase and preventing activation of NF-kß, as well as by suppressing nuclear translocation of signaling pathway components
JAK/STAT	Cytokines (IL-2, IL-6, IL-10, IL-12, IFN-α, etc.) and growth factors (EGF)	Regulation of cellular proliferation, differentiation, and inflammatory response	Suppression of binding of transcription factor STAT3 to the promoter (ATF6)
PPAR-α/γ	Unsaturated fatty acids, eicosanoids, etc.	Regulation of metabolism of fatty acids and energy balance, regulation of immune response, and cellular cycle	Competitive interaction due to binding to RXR and suppression of the PPAR-α/γ promoter
NFAT	Ca2^+^	Regulation of cellular cycle and inflammatory response	Blocking of NFAT components of Runx1 transcription factor

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
