# Peer review of "Cellular and Molecular Genetic Mechanisms of Lung Fibrosis Development and the Role of Vitamin D: A Review"

_ijms, 2024, doi:10.3390/ijms25168946_

Round 1

Reviewer 1 Report

Comments and Suggestions for Authors

Enzel and her co-authors' manuscript is dedicated to the potential use of Vitamin D for pulmonary fibrosis. The literature analysis of the data is very superficial, there is not any experimental evidence of the effectiveness of Vitamin D against pulmonary fibrosis. There is no particular pharmacological target suggested in the review. The author's statement, that profibrotic inflammatory response is a key target is too wide and nonspecific. Indeed, Vitamin D can affect the expression of cytokines and chemokines, involved in fibrogenesis, as other vitamins or antioxidants, and slightly modulate the immune response. However, there is no link between the mechanism of Vitamin D and extracellular matrix formation.  

Comments on the Quality of English Language

The language needs a major revision. The text looks like the manuscript was written in the author's language of origin and translated using software. 

Author Response

The authors are grateful for the analysis of the submitted manuscript. The purpose of this review is to highlight key current concepts regarding the pathogenesis of pulmonary fibrosis as well as potential targets of vitamin D and its derivatives. The authors believe that the review article, in addition to summarizing the available information, should have problems, identify unresolved issues, and encourage further research. The authors agree that the data available and presented in the manuscript regarding preclinical and clinical studies on the use of vitamin D (or synthetic vitamin D receptor agonists) are often contradictory. At the same time, an attempt was made to explain this contradiction. Recently, convincing evidence has emerged regarding the potential application and potential beneficial effects of vitamin D in experimental models of pulmonary fibrosis. At the same time, possible limitations in extrapolating data to humans include the imperfection of the experimental model of bleomycin-induced fibrosis, which is highlighted in the article. The authors believe that the data presented in the review will allow readers to generalize and systematize existing ideas and, possibly, initiate further experimental and clinical research in this area.

General notes:

The entire manuscript has been carefully checked for the correct English language. All necessary edits have been made to the text.

Reviewer 2 Report

Comments and Suggestions for Authors

In the current manuscript the authors have reviewed the recent understanding of the mechanisms of pulmonary fibrosis (PF) development in context of the critical roles played by vitamin D in this situation. Vit D has pleiotropic roles in relation to cell proliferation, differentiation, apoptosis, intercellular adhesion, oxidative stress, matrix homeostasis, and regulation of inflammatory responses. The current literature review summarizes the latest understanding of pulmonary fibrosis in terms of potential application points of effects of ligand-associated activation of vitamin D receptors, as well as to present existing experimental and clinical data on the effects of vitamin D and its analogues in conditions of progressive pulmonary fibrosis. Overall this is a well-conceived and written manuscript. I have a few suggestions to improve the content.

1.       The manuscript needs to be concise and focused. Any ideas discussed in each section should include a summary statement at the beginning to orient the readers on why the pathway or mechanisms is/are important and what is/are the significance of them in this particular situation.

2.       The manuscript mentions clinical trial (CT)-based approaches to inhibit TGF-β for controlling PF. Similarly clinical trials targeting suppression of other growth factors, including vascular endothelial growth factors 46 (VEGFR-1, VEGFR-2 and VEGFR-3), fibroblast growth factors (FGFR-1, FGFR-2 and 47 FGFR-3) and platelet growth factors (PDGRF-αand PDGRF-β) have also been mentioned. It would be useful to construct a table showing drug, target, CT number, phase of CT, nature of the study (placebo-controlled randomized etc), nature of enrolling cases (prospective vs retrospective, longitudinal vs cross sectional) and outcome of the study results if available.

3.       The manuscript provides us with data from a large scale randomized CT (p12/21) indicating ‘absence of any significant links between genetically determined levels of circulating vitamin D (25(OH)D) and the risk of developing idiopathic pulmonary fibrosis’ which certainly underestimate the relevance of vit D in the present context. The authors need to emphasize with plausible explanation(s) for this apparent discrepant results.

4.       The manuscript lacks detailed descriptive self-explanatory figure legends.

5.       Include a table listing the abbreviations and the full name. This will help reach a larger readers from the concerned and allied field of research.

Minor:

1.       All in vitro and in vivo used throughout the manuscript need to be italicized.

2.   Provide detailed information of the tools/soft wares used in generating figures.

Author Response

We are very grateful to you and all the reviewers for your time and work, your suggestions and recommendations for improving the submitted manuscript! The authors appreciate the suggestions expressed by the reviewers with great respect and attention, and for the most part, all the suggestions were reflected within the corrected manuscript. The changes made to the manuscript are highlighted in green.

Response to comments and suggestions from Reviewer #1:

  1. The manuscript needs to be concise and focused. Any ideas discussed in each section should include a summary statement at the beginning to orient the readers on why the pathway or mechanisms is/are important and what is/are the significance of them in this particular situation. – Done. At the beginning of each subsection of the article, a brief summary is added with an emphasis on the issue under consideration, without duplicating the main content of the subsection.
  2. The manuscript mentions clinical trial (CT)-based approaches to inhibit TGF-β for controlling PF. Similarly clinical trials targeting suppression of other growth factors, including vascular endothelial growth factors 46 (VEGFR-1, VEGFR-2 and VEGFR-3), fibroblast growth factors (FGFR-1, FGFR-2 and 47 FGFR-3) and platelet growth factors (PDGRF-αand PDGRF-β) have also been mentioned. It would be useful to construct a table showing drug, target, CT number, phase of CT, nature of the study (placebo-controlled randomized etc), nature of enrolling cases (prospective vs retrospective, longitudinal vs cross sectional) and outcome of the study results if available. – Done. At the beginning of the manuscript, a table with a brief description of the key clinical trials and the results obtained is added. Unfortunately, the scope of this review does not allow us to cover all the clinical trials with IPF patients, but the information presented, in the authors' opinion, allows for a general impression of the key areas of the search for therapeutic approaches, with an emphasis on the pathogenetic mechanisms of pulmonary fibrosis development.
  3. The manuscript provides us with data from a large scale randomized CT (p12/21) indicating ‘absence of any significant links between genetically determined levels of circulating vitamin D (25(OH)D) and the risk of developing idiopathic pulmonary fibrosis’ which certainly underestimate the relevance of vit D in the present context. The authors need to emphasize with plausible explanation(s) for this apparent discrepant results. – Done. The authors agree that the presented results of the above-mentioned study are contradictory with the existing fundamental data regarding the potential effects of vitamin D. At the same time, the article also discusses the problematic discrepancy between the results obtained in preclinical and clinical studies. As a possible explanation for the contradictory results, information regarding diet-gene or gene-environment interactions, as well as the concept of the personal vitamin D response index, was added to the manuscript.
  4. The manuscript lacks detailed descriptive self-explanatory figure legends. – Done. Both figures are now accompanied by short, self-explanatory legends.
  5. Include a table listing the abbreviations and the full name. This will help reach a larger readers from the concerned and allied field of research. – Done. At the end of the article, a table with abbreviations has been added (if it is permitted by the journal rules).

Minor:

All in vitro and in vivo used throughout the manuscript need to be italicized. – Done. All mentions related to in vitro and in vivo studies are provided in italics now.

Provide detailed information of the tools/soft wares used in generating figures. – Done. Figure legends now state – The scheme was generated using images from Servier Medical Art. Servier Medical Art by Servier is licensed under a Creative Commons Attribution 3.0 Unported License (https://creativecommons.org/licenses/by/3.0/).

Reviewer 3 Report

Comments and Suggestions for Authors

Nicely written review. Author may include to enhance the quality of the review.

1. The author may add a demographic table for the incidences of lung fibrosis, including country wise distribution, yearly progression of the case of lung fibrosis etc.

2. Author may also include a comparative graph to show distribution of lung fibrosis according to age, sex, smoking etc.

Author Response

  1. The author may add a demographic table for the incidences of lung fibrosis, including country wise distribution, yearly progression of the case of lung fibrosis etc.
  2. Author may also include a comparative graph to show distribution of lung fibrosis according to age, sex, smoking etc.

The authors are grateful for the suggestions and believe that the emphasis on epidemiological data regarding the occurrence of idiopathic pulmonary fibrosis is a necessary component indicating the relevance of searching for new, highly effective approaches to treat the patients and further studying the molecular pathogenesis of fibrotic transformation. At the same time, the authors believe that the scope of this review article does not allow for full coverage of the epidemiological data available in the literature. Instead of forming an additional table and graph, the authors considered it possible to present expanded epidemiological data indicating the prevalence of the disease in various regions, a growing trend in morbidity, as well as greater susceptibility among elderly people with a predominance of males and smokers.

Round 2

Reviewer 2 Report

Comments and Suggestions for Authors

The authors have taken great care in addressing my comments and concerns. The point-by-point responses provided by the authors are satisfactory and the revised version of the manuscript appears much improved.